# The Use of Human Sterilized Crushed Tooth Particles Compared with BTCP Biomaterial and Empty Defects in Bone Formation inside Critical Rabbit Calvaria Sites

**DOI:** 10.3390/bioengineering10060638

**Published:** 2023-05-24

**Authors:** José Luis Calvo-Guirado, Marta Belén Cabo-Pastor, Francisco Martínez-Martínez, Miguel Ángel Garcés-Villalá, Félix de Carlos-Villafranca, Nuria García-Carrillo, Manuel Fernández-Domínguez

**Affiliations:** 1Health Sciences Faculty, Universidad Autónoma de Chile, Santiago de Chile 7500912, Chile; 2Dentistry Department, Universidad CEU Cardenal Herrera, 46001 Valencia, Spain; marta_belen.cabo@uchceu.es; 3Department of Orthopaedic Surgery and Traumatology, Hospital Clínico Universitario Virgen de la Arrixaca, 30120 Murcia, Spain; fmtnez@gmail.com; 4Department of Implant and Biomaterial Research, Fundación Corazón de Jesús, San Juan 5400, Argentina; miguelgarcesodont@gmail.com; 5Department of Orthodontics, Faculty of Medicine, University of Oviedo, 33001 Asturias, Spain; decarlosfelix@gmail.com; 6Department of Veterinary, University of Murcia, 30007 Murcia, Spain; ngc51078@gmail.com; 7Department of Dentistry, Universidad Camilo José Cela, 28001 Madrid, Spain; clinferfun@yahoo.es

**Keywords:** bone graft, bone substitutes, dentin grinder, skull rabbits, tooth particles, BTCP biomaterial, crushed teeth, human sterilized crushed teeth

## Abstract

This study aimed to assess the bone regeneration of critical-size defects in rabbit calvaria filled with freshly crushed extracted teeth, comparing them with BTCP biomaterial and empty sites. Materials and methods: Twenty-one female New Zealand rabbits were used in this study. Two critical-size defects 6 mm in size were created in the skull bone, each with a 3 mm separation between them. Three experimental groups were evaluated: Group A (human sterilized crushed teeth granules alone), Group B (Bioner Bone, Bioner Sitemas Implantológicos), and Group C (unfilled defects). The animals were sacrificed at 4 and 8 weeks. Evaluation of the samples involved histological and histomorphometric analyses with radiographic evaluation. The histological evaluation showed a higher volume reduction in Group A compared with Group B (*p* < 0.05) and Control. Group A showed the highest values for cortical closure and bone formation around the particles, followed by Group B and Group C (*p* < 0.05). Within the limitations of this animal study, we can conclude that the use of human tooth particles leads to increased bone formation and reduced connective tissue in critical-size defects in rabbit calvaria when compared to BTCP biomaterial. The calvarial model is a robust base for the evaluation of different biomaterials.

## 1. Introduction

Many bone defects can heal on their own, but the resorption process is more significant; therefore, it is necessary to take care of the bone walls using biomaterials [1]. The use of collagen membranes is intended to seal the soft tissue and allows the growth of hard tissue underneath it without the formation of connective tissue [2]. In any cases, part of this regeneration process is not achieved because the defect is critical, is greater than 6 mm in size, and bone formation is not obtained correctly, leading to increased bone resorption. Additionally, this can result in a lengthened postoperative period before the implants can be placed [3]. Among the most important alternatives for bone regeneration, we have allograft materials and synthetic bone. Now, the granules of teeth are incorporated, which provide ideal characteristics for the regeneration of bone [4]. Bone filling materials have mechanical properties, high porosity, and high resorption. Above all, the composition associated with bone can be modified according to some clinical requirements and the defect [4]. Beta tricalcium phosphate (BTCP) is one of the most active forms of plaster of Paris and is part of this group with a high reabsorption capacity. Many of the ceramic lips and their superiors are not resorbed in the bone and remain within the defect for an extended period, serving as scaffolding, allowing bone regeneration, and disappearing in the long term over the course of ten years [5]. Regarding teeth, many extracted teeth are discarded from our clinics. However, we can give them a second chance by utilizing them as a biomaterial. Additionally, the patient’s own extracted teeth can be used immediately or after implant placement surgeries or partner filling. Our findings suggest that a crushed tooth compared to beta-tricalcium phosphate has better characteristics for osteogenesis and allows the dentin to act as an osteoinductive material [6]. Some authors have experimented with biomaterials in critical defects of more than 7 mm in rabbits’ skulls, allowing rapid bone regeneration using demineralized bovine bone. Compared to these results with beta-tricalcium phosphate, mineralized bovine bone allows for a much better and more stable scaffold compared to beta-tricalcium phosphate, which is rapidly absorbed and resorbed [7,8]. The addition of silicate to the biomaterial beta-tricalcium phosphate activates it and improves the stability of the biomaterial, providing it with structural and functional properties to improve bone formation in small animals using porous granular hydroxyapatite as a scaffold [9]. Another biomaterial that has been incorporated within the field of biomaterials is granulated teeth. However, some authors use the tooth as an online graft with resolution characteristics like human bone and with a resorption process that may last up to 12 weeks [10]. Animal studies reveal promising results with this technique, where the particulate granulated tooth is placed in the post-extraction alveoli; thus, promoting immediate bone regeneration [11]. The new machine for grinding teeth is called Smart Dentin Grinder Genesis (Kometabio NJ, EEUU) and has been designed to be able to grind, crush, and lift the tooth particles in two different compartments. It is an instrument that facilitates the reuse of extracted teeth as a new biomaterial by applying a dentin cleanser for 5 min and a buffer for 30 s, which can remove bacteria from the freshly crushed tooth and clean the particles from the crushed tooth [12]. This particulate tooth is available in two sizes of particles (300 and 1200 microns) and can be used for the preservation of the post-extraction alveolus, as well as in temporary implants and especially in immediate loads. The use of dentin is essential for its placement in post-extraction alveoli due to its osteoinductive capacity. Enamel must be placed on the vestibular wall to regenerate idle defects in the anterior sectors [13]. The amount of biomaterial that we can obtain from the extracted teeth allows us not only to fill the alveoli in quantity and quality, but also to preserve the bone structures and improve the stability of the bone, as well as its bone density [14,15,16,17,18]. The use of the tooth as a graft has been shown to have benefits over other biomaterials due to the presence of collagen and growth factors and, above all, due to its bone-like structure. However, a significant discrepancy among different clinical studies is that the resorption time of the granulated tooth varies between 2 and 24 weeks, with a risk of dehiscence that ranges from 12 to 34% over time [19]. This study aims to compare two different biomaterials, such as human sterilized disinfected particulate tooth graft and beta-tricalcium phosphate, in 6 mm critical defects in New Zealand rabbit calvaria to evaluate new bone formation and closure of the bone defect within a time of four to eight weeks.

## 2. Materials and Methods

Biomaterials can regenerate bone defects by stimulating bone cells.

The specific objectives were:

To assess which biomaterial (dentin graft or BTCP) reabsorbs faster.

To assess which biomaterial forms more bone at 4 and 8 weeks.

The other secondary objectives were to determine which biomaterial can regenerate critical 6 mm bone defects by stimulating bone cells and promoting bone formation.

### 2.1. Materials

#### 2.1.1. Animals

Twenty-one male New Zealand rabbits aged 31–35 weeks weighing 3.8–4.2 g were included in the study. The research study was approved by the ethics committee of the University of Murcia REGA ES300305440012, Spain, with the approval code A1320140404, which followed guidelines established by the Council Directive of the European Union (53/2013; 1 February 2013) for animal care and experimentation. The animals used in the experiment were fed a soft diet with water ad libitum throughout the study period, which was controlled by the veterinary service of the University of Murcia. Rabbit skulls are helpful in the scientific world due to similarities between the left and right parts of the parietal bones of the animal’s head. Calvaria bone is essential due to many blood vessels present in this bone, which is approximately 1 to 2 mm thick, and its cortical composition allows the assessment of new bone formation over a short period of time. New bone formation mainly occurs due to the blood supply that is present in the bone and, above all, allows the assessment of the density and quality of the obtained bone [20]. Furthermore, the use of the skull as a test for biomaterials is justified due to its similarity with the maxillofacial regions of the human being, stemming from their shared embryological origin, where the bones of the face developed from a membranous precursor [21].

#### 2.1.2. Biomaterials

One of the biomaterials used was Bioner Bone (Bioner Sistemas Implantológicos, Barcelona, Spain), which is a synthetic material with a high resorption capacity that enables nice and stable bone regeneration. This material comprises an entirely synthetic bone graft material composed of 100% beta-tricalcium phosphate and features 80% interconnected microporosity and macroporosity. This biomaterial has a granule size ranging from 0.5 to 1 mm particle thickness (Figure 1).

#### 2.1.3. Crushed Teeth

The study protocol was approved by the Catholic University of Murcia Ethics Committee related to the use of extracted teeth for in-vitro studies (UCAM; registration number 6781; 21-07-2017). Mandibular anterior human teeth were extracted from 10 patients aged between 50 and 60 years, with an S.D. of (55.34 ± 0.16), including seven men and three women. All the patients signed informed consent forms to donate their teeth for the study and did not receive financial compensation. The patients’ teeth were diagnosed with periodontal disease in the lower six anterior teeth (numbers 31 to 41). The inclusion criteria were patients over 50 years of age and systemically healthy patients who needed an anterior mandibular tooth/teeth extraction for periodontal reasons. All the teeth were healthy with a mobility grade of 3 and were without infection. The exclusion criteria included patients who do not need an anterior mandibular tooth extraction, systemically complicated patients (ASA III, IV, V), pregnant or lactating patients, and patients who were not willing to complete follow-up calls. The inclusion and exclusion criteria were described in a previous article by Calvo Guirado et al. [14]. After the extracted human teeth were cut, only the root was used. The roots were cleaned with a carbide tungsten bur. Then, a dental chair syringe was used to dry the roots and the roots were ground using a designed ‘Smart Dentin Grinder Genesis’ device. The 300–1200 μm dentin particulate was sieved through a particular sorting system (Figure 2). The standard used for the number of dental particles implanted during surgery was a particle thickness of 0.5–1 mm.

The particles of 1200 microns in size were then immersed in a unique basic alcohol dentin cleanser (red cap) in a sterile container to dissolve all the organic debris and bacteria for 5 min at room temperature. Then, the particulate dentin was treated with EDTA (ethylene diamine tetra acetate, blue cap) for 2 min. This step was used for the partial demineralization of tooth particles. Finally, the cleaned particles were washed twice with PBS (phosphate-buffered saline, green cap) solution for 5 min to neutralize the pH levels (Figure 3).

The scanning electron microscopy of crushed human tooth particles used as a graft demonstrated large amounts of collagen. They also have a higher porosity, especially after treatment with the liquids recommended by the manufacturer, which would favor the entry of blood vessels (Figure 4).

After this procedure, the human crushed dentin particles were sterilized by autoclave, and then the bacteria-free particulate tooth graft was grafted into the calvaria defects of rabbits, which were randomly selected using the www.randomization.com (accessed on 21 April 2023) program. Based on the assumptions made, research has verified the efficacy of the cleanser about bacterial elimination, which was observed through the destruction and inhibition of existing microorganisms, as outlined in the primary protocols [12].

### 2.2. Study Design

Forty-two 6 mm critical-size defects located in both sides of the rabbits’ parietal bones were performed in 21 New Zealand male rabbits. The development of this study of critical effects allows the rabbit to be used as an experimental model for rapid and effective bone regeneration. The critical effects are difficult to achieve, but these experimental animals allow us to regenerate bone more quickly and allow us to evaluate the outcomes within shorter periods of time compared to human bone regeneration. Furthermore, this in vivo experimental model also allows the evaluation of the activity of two different biomaterials, one natural and the other chemical, in promoting new bone formation and allowing adequate resorption.

The research was divided into three experimental groups (Figure 1):

Group A: 14 critical-size defects filled with BTCP (Bioner Bone)

Group B: 14 critical-size defects filled with human sterilized crushed teeth.

Group C: 14 critical-size defects were left unfilled as controls.

### 2.3. Surgical Procedure

The animals were prepared to receive an intramuscular injection of 0.5 to 1 mg/kg acepromazine maleate. Fifteen minutes later, a general anesthesia of 5 to 8 mg/kg ketamine plus chlorbutol was administered intravenously with 0.05 mg/kg atropine as a coadjuvant. The rabbits’ skulls were shaved, washed, and disinfected with Sea4 Encías (Blue Sea Laboratories, Alicante, Spain), and an anteroposterior midline skin incision was made. Full-thickness flaps were raised, exposing the parietal bone, and two 6 mm circular bilateral bicortical defects were created with a 6 mm stainless steel trephine bur at 800 RPM (Figure 5). After a randomization test, each defect was filled according to its test group allocation (Group A, Group B, and Group C). Collagen membranes were used to cover the defect. The periosteal and subcutaneous tissue layers and skin were then closed with vicryl 3/0 and 3/0 silk sutures, respectively. Postoperative pain and distress were controlled by subcutaneous buprenorphine at a dose of 0.02 mg/kg every 8–12 h for 3 d.

### 2.4. Animal Sacrifice

The animals used in the study were sacrificed at four (10 animals) and eight (11 animals) weeks after the initial surgery.

### 2.5. Sample Processing

The specimens were dehydrated in a graded ethanol series (70, 90, 95%, and absolute ethanol), after which the specimens were infiltrated with Technovit 7200 VLC light curing resin (Kulzer, Wehrheim, Germany) for 21 days. Then, the samples were embedded, and light polymerized with an EXAKT 520 polymerization system (EXAKT Technologies, Oklahoma City, OK, USA). Finally, coronal sections were cut using the EXAKT 310 CP cutting unit (EXAKT Technologies, Oklahoma City, OK, USA). All the sections that were obtained after cutting were approximately 200 µm in thickness and were manually polished to a final thickness of 35–50 µm. The samples were stained with picrosirius hematoxylin staining. This procedure was previously published with my group [14,15,16].

### 2.6. Histomorphometric Study

The prepared samples were photographed with a digital camera using a 20×motorized optical microscope (BX51, Olympus, Japan). These photographs were combined using a computer program (Sense Cell Dimensions, Olympus, Japan) to obtain high-resolution images of the entire sample. The regions of interest (R.O.I.) were manually delimited to facilitate the identification of the different tissues present in each sample. This procedure was previously published with my group [14,15,16].

The following variables were recorded at two study times (4 and 8 weeks):

New Bone Formation (N.B.F.): the percentage of new bone inside the marrow space and between the P.T.G. particles in the R.O.I.

Residual Graft Material (R.G.M.): the percentage of particles present inside the R.O.I. concerning the total area.

Connective Tissue (C.T.): connective tissue or the connective tissue space inside the R.O.I. expressed as a percentage.

### 2.7. Statistical Analysis

The statistical data were produced using SPSS 21.0. software (SPSS Inc., Chicago, IL, USA). After the descriptive analysis, the Kolmogorov–Smirnov test was used to evaluate the significance of the differences between two different 6 mm holes in Group A (Bioner Bone) and Group B (human sterilized crushed teeth). A two-way ANOVA for paired samples was used to evaluate differences between the volumetric densities among the two grafting materials and the original calvaria bone. Statistical significance was set at a value of *p* < 0.05. A two-sample *t*-test was used to compare the volumetric densities of tissues for the exact bone substitute between Group A and Group B. The significance level was set at *p* ≤ 0.05 [14,15,16].

## 3. Results

No type of inflammation, infection, or loss of biomaterial was found throughout the study at four and eight weeks.

### 3.1. Radiovisiography

Figure 6 and Figure 7 show the condensation of more homogeneous dentin grafts in rabbits’ skulls at four and eight weeks compared with BTCP ^®^. 

### 3.2. Histomorphometric Analysis

The evaluation of bone neoformation was divided into two parts: one part closest to the native bone, called the region of interest 1 (R.O.I. 1), and the graft area itself, called the region of interest 2 (R.O.I. 2) (Figure 1).

### 3.3. Histologic Evaluation

#### Four Weeks

Human sterilized crushed teeth group (Group B) showed that, around the old bone (R.O.I. 1) and the region attached to the crushed human tooth, a more significant amount of immature bone in the bone defect was created. Furthermore, the presence of osteogenic connective tissue in the remodeling phase was demonstrated. In the central part of the area of interest grafted by a crushed tooth (R.O.I. 2), we observed that the bone had more significant maturation in the center of the defect, with very well-calcified edges surrounding it (Figure 8). Bioner Bone ^®^ (Group A) showed slightly new trabeculae bone around the region of interest of the old bone, which was part of the central defect that we had created (R.O.I. 1). This new bone tissue contains many capillaries and marrow spaces that have increased the thickness of the central area (R.O.I. 2) of the defect compared to the pre-existing area of old bone (Figure 8). For the control group (Group C), no bone formation was observed (Figure 9).

At eight weeks, the human sterilized crushed teeth group (Group B) showed the formation of enormous trabecular tissue with wide medullary spaces within the bone defect, which favored the maturation of the leisure tissue in the defect (Figure 10). The trabecular bone presents thick and well-defined mature characteristics with characteristics of advanced calcification around the grafted area. For the BTCP biomaterial group (Group A), the defect had increased in thickness with new trabeculae bone formation, high blood supply, and partially healed bone-grafted areas (Figure 11). For the control group (Group C), no bone formation was observed.

### 3.4. Histomorphometric Study

#### 3.4.1. Four Weeks

Cortical Defect Closure (C.D.C.): All the groups showed great approximation to the edges of the old bone of the calvarium, where a reduction in the size of the original defect of 6 mm was observed. Group B (dentin graft) showed high bone formation (51.4% ± 0.7%) with complete closure of the bone defect. A mixture of medullary bone surrounding the tooth particles contributed to the formation of a new cortex. In group A, there was a partial closure of the defect (20.6% ± 0.18%) of 6 mm, also compared with the control group (11.2% ± 0.2%), where there was no complete healing of the initial defect.

Residual Graft Material (R.G.M.): The most residual biomaterial was found in Group B, corresponding to the crushed teeth, which was more than in Group A and the control group since, in the latter, there was no biomaterial included within the defect (*p* < 0.05).

Connective Tissue (C.T.): It was to be expected that the control group (88.8% ± 0.7%) would be where more connective tissue would be found since it was not filled with any suitable material, followed by Group A and finally by Group B, where the connective tissue had an osteogenic origin but was found in lesser quantity than the newly formed bone (*p* < 0.05) (Table 1).

#### 3.4.2. Eight Weeks

Cortical Defect Closure (C.D.C.): All the groups showed a reduction in the size of the defect compared to those found at four weeks. Group B showed the best closure of the bone defect with a mature bone quality (61.3% ± 1.6%), compared to group A, which also healed at eight weeks; however, the bone was of lower quality and quantity (38.2% ± 0.11%). The control group (18.3% ± 0.17%) did not show complete closure of the defect (*p* < 0.05).

Residual Graft Material (R.G.M.): Related to the percentage of persistent biomaterial, Group B had the most residual material due to the high crystallinity of the tooth (21.9 ± 1.5%), compared to group A, which was grafted with a highly resorbable material (15.6 ± 0.2%). The most important thing about this detail is that only 21% of the tooth remained due to its high resorption, and 15% of the biomaterial was BTCP. The control group had no record of residual material (*p* < 0.05).

Connective Tissue (C.T.): The control group (70.61% ± 3.7%) was found to be the group with the highest percentage of connective tissue, followed by Group A and Group B (*p* < 0.05) (Table 2).

### 3.5. Laser Optical Microscopy

The use of the Keyence Digital microscope VHX-7000 (Keyence Corporation, Mechelen, Belgium) enabled the observation that, after four weeks, there was more prominent mild bone neoformation at R.O.I. 2 in the dentin graft concerning the biomaterial used in the study. At eight weeks, the group treated with crushed teeth (Group B) showed images of well-organized and stable new bone formation with great vascularity and cellularity where osteons can be observed, indicating complete bone regeneration (Figure 12, Figure 13 and Figure 14).

## 4. Discussion

For many years, teeth have been used to regenerate bone, as has been described in several studies carried out by Urist, taking advantage of the possibility of regenerating bone by applying mineralized teeth. We all know that the tooth is ideal for reconstructing some defects, both vertical and horizontal, due to its bone formation capacity and its composition. The composition of the tooth with type I collagen provides it with unique characteristics for osseointegration and acts as a trigger for bone regeneration [21,22,23,24]. The use of dentin, due to its high compatibility with bone tissue, is capable of stimulating bone formation thoroughly, incorporating live bone with high vascularity within the bone defect. In addition, bovine dentin has osteoconductive and non-osteoinductive potential, similar to human dentin, which has been used to repair and promote the regeneration of bone tissue [25]. Our previous results have revealed that interactions between dentin and bone tissue cells allows a stable union, stimulating the formation of new bone due to the scaffolding capacity of the particulate tooth and favoring bone mineralization [14,15,16,17,18]. The main reason we have used the rabbit shell is that it is an experimental model proven for its effectiveness. Additionally, this biomaterial is able to promote osseo-conduction and osseo-induction in bone structures with minimum thicknesses, as has been revealed in different experimental studies [26]. After the extractions, the graft of the particulate tooth from the human being should be considered another biomaterial to fill the post-extraction alveoli, protecting the buccal and lingual wall. In this way, it allows the regeneration of the medullary bone at 60 days and the lamellar bone at 90 days of healing, favoring the maintenance of the bone walls [24]. Our previous results have revealed that the interaction of dentin with bone tissue cells allows a stable union, stimulating the formation of new bone due to the scaffolding capacity of the particulate tooth and stimulates bone mineralization [14,15,16,17,18]. The graft of the particulate tooth from the human being after the extractions should be considered another biomaterial to fill the post-extraction alveoli, protecting the buccal and lingual wall. In this way, it allows the regeneration of the medullary bone at 60 days and the lamellar bone at 90 days of healing, favoring the maintenance of the bone walls [24]. Our results coincide with other authors’ results that the crushed tooth has the capacity to maintain the space of four walls, prevent the crestal resorption of the bone walls, and regenerate through the collagen of the crushed tooth, forming new bone [15]. Other authors reveal that the demineralization process leads to an increase in BMP-2, and what makes this bone development more effective is that it acts as an osseo conductor, similar to other biomaterials that regenerate bone [27]. Indeed, when sterilizing the crushed tooth from the human being, part of the collagen used to regenerate the bone is lost, but hydroxyapatite remains as a scaffold for forming new bone [28]. All these findings ultimately show that the human-derived crushed tooth graft PR-DDM can actively contribute to bone formation by acting as a scaffold for the vascularization of that tissue [29]. Other authors make a mixture of the crushed tooth with beta-tricalcium phosphate, thus improving osteogenesis by favoring the initial reabsorption of beta-tricalcium phosphate, maintaining the tooth as a scaffold [30]. Another way to favor bone regeneration and its increase is to increase the compressive forces applied to the bone particle in minor defects created in the rabbit’s skull. This reconstruction with different biomaterials quickly obtains optimal revascularization and bone regeneration [31]. Finally, we can say that using sterilized human crushed teeth stimulates bone formation in critical defects that do not heal by themselves without interfering in the regeneration process, acting as a solid material that can repair damaged bone.

## 5. Conclusions

Despite the limitations of this rabbit study, we conclude that the use of human sterilized crushed teeth showed better new bone formation and allows critical-size defects to heal without interfering in the regeneration process, compared with the BTCP group and unfilled areas. Human sterilized crushed teeth is a suitable material for healing critical-size defects and also maintains a scaffold for new bone formation.

## Figures and Tables

**Figure 1 bioengineering-10-00638-f001:**
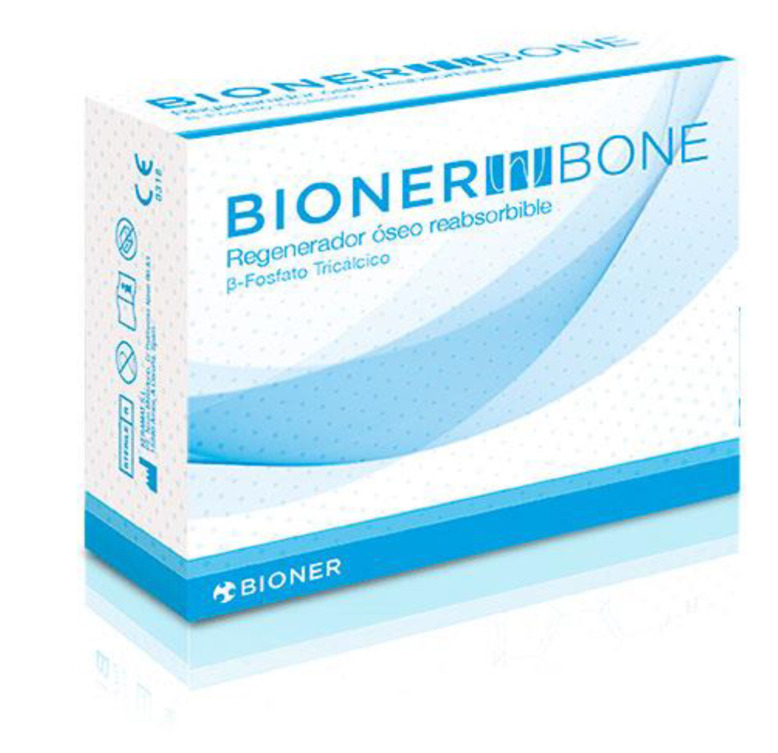
Bioner bone^®^ (BTCP biomaterial).

**Figure 2 bioengineering-10-00638-f002:**
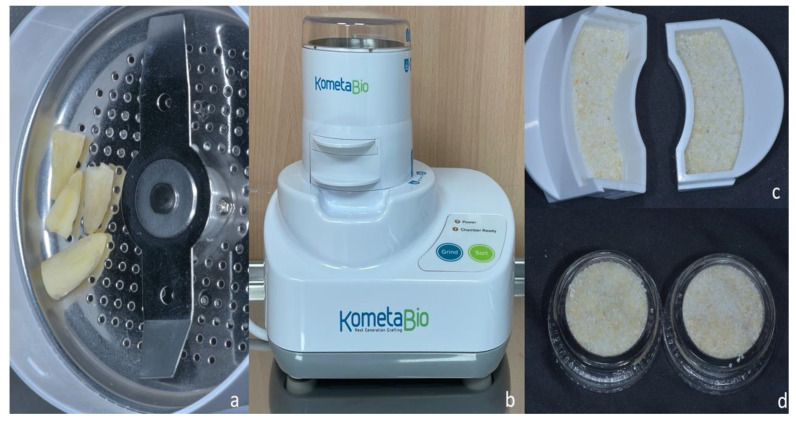
(**a**) Four extracted roots inside the grinding chamber after teeth cleaning; (**b**) dentin graft machine (Smart Dentin Grinder Genesis, Kometabio, NJ, EEUU); (**c**) two ground particle compartments; (**d**) crushed teeth in glass recipients.

**Figure 3 bioengineering-10-00638-f003:**
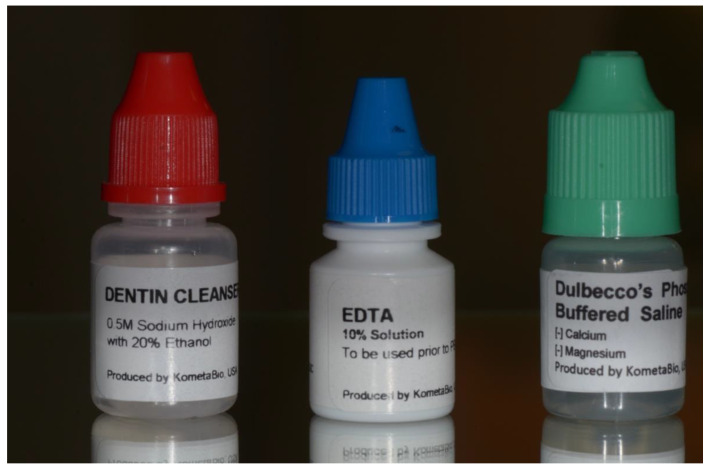
Disinfection liquid solutions: unique basic alcohol dentin cleanser (red cap), EDTA (ethylene diamine tetra acetate, blue cap), and PBS (phosphate-buffered saline, green cap).

**Figure 4 bioengineering-10-00638-f004:**
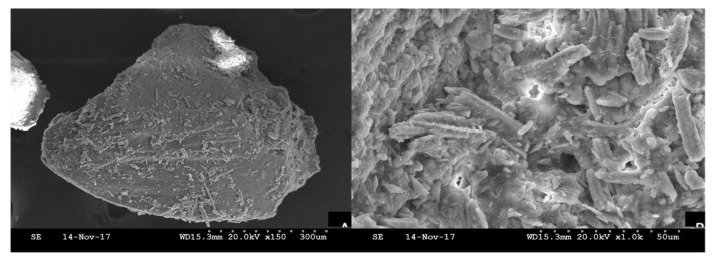
Scanning electron microscopy (S.E.M.) of human crushed sterilized tooth particles.

**Figure 5 bioengineering-10-00638-f005:**
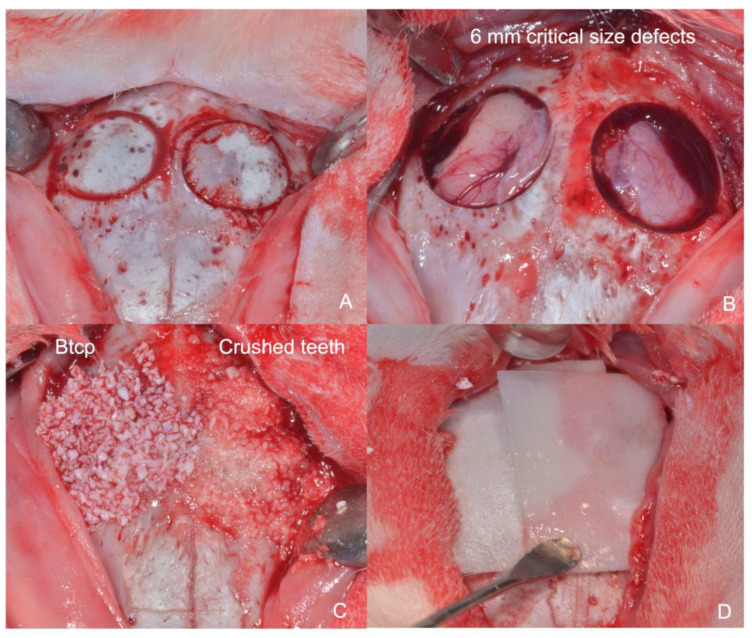
(**A**): Trephine critical cylindrical defects in both parietal bones; (**B**): 6 mm critical-size defects; (**C**): BTCP was placed on the right side and human sterilized crushed teeth was placed on the left side; (**D**): collagen membrane covering both biomaterials used in the study.

**Figure 6 bioengineering-10-00638-f006:**
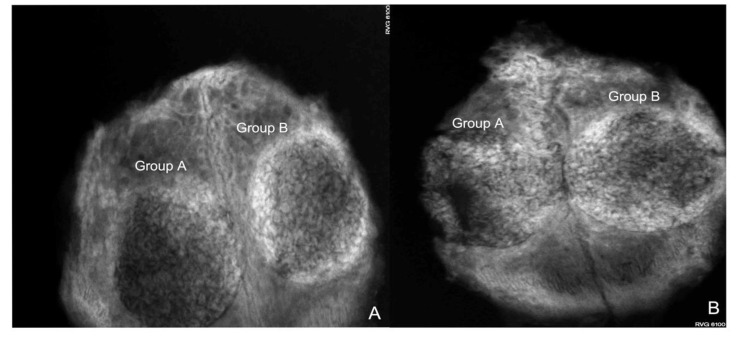
(**A**) Digital image of the rabbit skull showing high bone formation around the human sterilized crushed teeth graft (Group B) and less bone formation in the group at four weeks of evaluation; (**B**) at eight weeks, the dentin graft group showed a circular calcified area compared with Group A, which showed slight bone healing.

**Figure 7 bioengineering-10-00638-f007:**
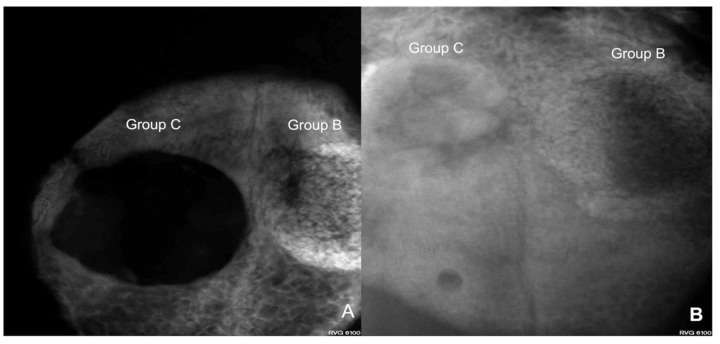
(**A**) Digital image of a rabbit skull showing more bone formation around the human sterilized crushed teeth graft (Group B) compared with the control group (Group C) at four weeks of evaluation; (**B**) at eight weeks, the dentin graft group (Group B) showed a lateral calcified area compared with slightly spontaneous bone healing in Group C.

**Figure 8 bioengineering-10-00638-f008:**
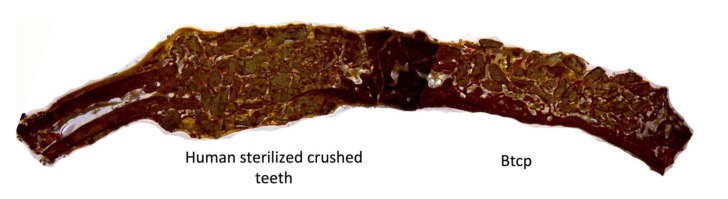
More bone formation in the area containing human sterilized crushed teeth compared with BTCP biomaterial at four weeks of evaluation.

**Figure 9 bioengineering-10-00638-f009:**
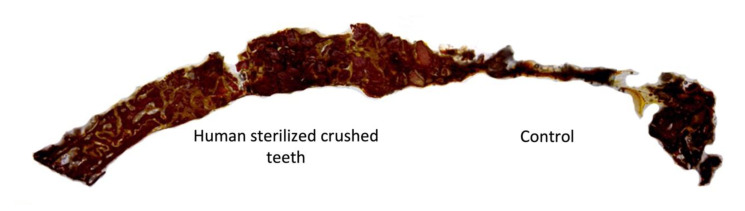
Bone formation in the area containing human sterilized crushed teeth compared with unfilled sites at four weeks of evaluation.

**Figure 10 bioengineering-10-00638-f010:**
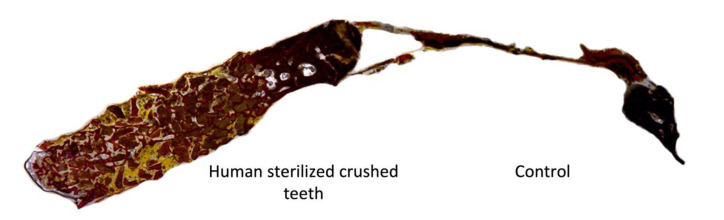
More bone formation in the area containing human sterilized crushed teeth compared with control sites at eight weeks of evaluation.

**Figure 11 bioengineering-10-00638-f011:**
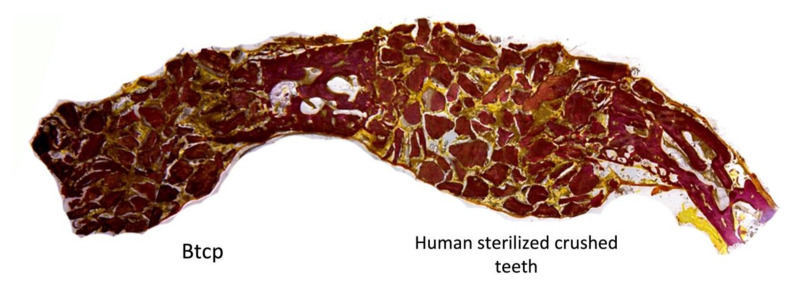
More bone formation in the area containing human sterilized crushed teeth compared with BTCP at eight weeks of evaluation.

**Figure 12 bioengineering-10-00638-f012:**
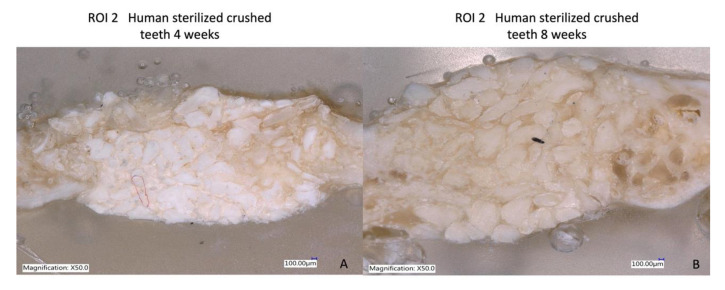
(**A**) Bone formation with dentin graft at four weeks; (**B**) more regenerated bone with tooth particles inside at eight weeks of follow-up.

**Figure 13 bioengineering-10-00638-f013:**
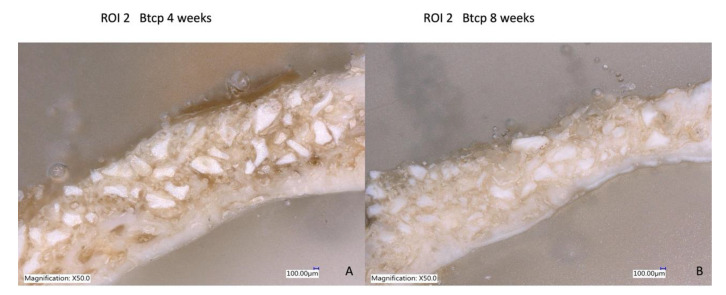
(**A**) Less bone formation with some particles at four weeks; (**B**) great resorption of BTCP at eight weeks of evaluation.

**Figure 14 bioengineering-10-00638-f014:**
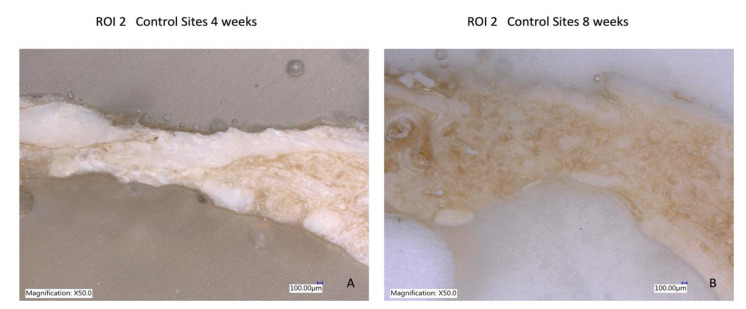
(**A**) No bone formation in critical-size defects at four weeks; (**B**) highly resorbed areas at eight weeks of follow-up.

**Scheme 1 bioengineering-10-00638-sch001:**
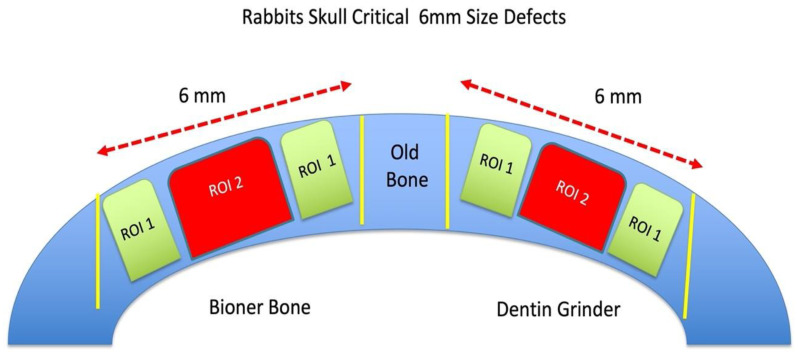
Design of the surgical procedure for rabbit skulls with two critical-size defects.

**Table 1 bioengineering-10-00638-t001:** Histomorphometric analysis of the percentage of new bone, residual material, and connective tissue at four weeks of follow-up. Data were expressed as percentages ± standard deviation. Group A (BTCP), Group B (human sterilized crushed teeth), and Group C (unfilled defect). Significance was set as *p* < 0.05 *.

GROUPS	Cortical Defect Closure (C.D.C.) %		Residual Graft Material (R.G.M.) %		Connective Tissue (C.T.) %	
	R.O.I. 1	R.O.I. 2	R.O.I. 1	R.O.I. 2	R.O.I. 1	R.O.I. 2
Group A	10.9% ± 1.2%	20.6% ± 0.18%	45.7% ± 10.6%	31.9% ± 0.2%	43.4% ± 0.8% *	47.6% ± 1.1% *
Group B	26.9% ± 0.6% *	51.4% ± 0.7% *	53.7% ± 1.3%	36.8% ± 0.5%	19.4% ± 1.2%	11.8% ± 0.7%
Group C	0.8% ± 1.1%	11.2% ± 0.2%	0.0% ± 0.0%	0.0% ± 0.0%	99.2% ± 0.3%	88.8% ± 0.7%

**Table 2 bioengineering-10-00638-t002:** Histomorphometric analysis of the percentage of new bone, residual material, and connective tissue at eight weeks of follow-up. Data were expressed as percentages ± standard deviation. Group A (BTCP), Group B (human sterilized crushed teeth), and Group C (unfilled defect). Significance was set as *p* < 0.05 *.

GROUPS	Cortical Defect Closure (C.D.C.) %		Residual Graft Material (R.G.M.) %		Connective Tissue (C.T.) %	
	R.O.I. 1	R.O.I. 2	R.O.I. 1	R.O.I. 2	R.O.I. 1	R.O.I. 2
Group A	27.6% ± 0.2%	38.2% ± 0.11%	21.3% ± 1.7%	15.6% ± 0.2%	51.1% ± 0.3% *	46.2% ± 1.6% *
Group B	31.5% ± 0.4% *	61.3% ± 1.6% *	41.8% ± 1.5%	21.9% ± 1.5%	26.7% ± 1.4%	16.8% ± 1.9%
Group C	11.2% ± 0.3%	18.3% ± 0.17%	0.0% ± 0.0%	0.0% ± 0.0%	88.8% ± 1.2%	81.7% ± 0.7%

## Data Availability

Data are contained within the article.

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
