# Peer review of "The Use of Human Sterilized Crushed Tooth Particles Compared with BTCP Biomaterial and Empty Defects in Bone Formation inside Critical Rabbit Calvaria Sites"

_bioengineering, 2023, doi:10.3390/bioengineering10060638_

Round 1
Reviewer 1 Report
Interesting manuscript comparing the utilization of a "traditional" grafting material vs. crushed teeth. The design of the investigation is good.
I would suggest for the writing to be revised to facilitate the reading.
-
Author Response
Dear reviewer
The paper was completely reviewed and corrected for better comprehension. Thanks a lot for your comments
All the best

Reviewer 2 Report
This study aimed to assess the bone regeneration of critical size defects in rabbit calvaria filled with freshly crushed extracted teeth compared with a BTCP biomaterial and empty sites. The author drew a conclusion through histological and histological econometric analysis that compared to BTCP biomaterials, human tooth particles increase the formation of bone defects in the critical size of the rabbit skull and reduce the formation of connective tissue. Too many similar studies have been reported, no novel information can be found, the innovation of this study seems insufficient. Below are some comments to the authors:
The rationality of the inclusion criteria for teeth collected in this study should be addressed, because the extraction of affected teeth due to periodontal disease might affect the conclusion.
Sterilizing the crushed dentin particles with autoclave usually influence the morphology of collagen in the teeth. Ultraviolet disinfection would be better.
The standard used for the amount of dental particles implanted during surgery is missing.
Is the implantation volume comparable to BTCP?
Fig. 1-3 should be deleted. Orther figures should be integrated into one figure.
Author Response
Comments and Suggestions for Authors
This study aimed to assess the bone regeneration of critical size defects in rabbit calvaria filled with freshly crushed extracted teeth compared with a BTCP biomaterial and empty sites. The author drew a conclusion through histological and histological econometric analysis that compared to BTCP biomaterials, human tooth particles increase the formation of bone defects in the critical size of the rabbit skull and reduce the formation of connective tissue. Too many similar studies have been reported, no novel information can be found, the innovation of this study seems insufficient.
1- Dear reviewer.
I completely agree with you with this comment, but let me tell you that this research was done in order to compare the crushed sterilized teeth with the most used biomaterial such as Btcp, and try to guess if these sterilized teeth act as a new biomaterial. Our paper differs from other immediately crushed teeth papers published by me and my group which were done with immediately extracted crushed teeth.
The other articles published by me were carried out on recently extracted periodontal teeth that preserved collagen, growth factors, and morphogenetic proteins. Our new paper is innovative since we can say that both recently extracted and sterilized teeth serve as biomaterial to fill bone defects.
It is also true, as you say, that the properties of collagen, growth factors, and morphogenetic proteins are lost when we sterilize crushed teeth, but we demonstrated that the tooth per se is a great independent bone filler material independently if the teeth were extracted immediately or extracted and sterilized.
Below are some comments from the authors:
2- The rationality of the inclusion criteria for teeth collected in this study should be addressed, because the extraction of affected teeth due to periodontal disease might affect the conclusion.
The inclusion criteria were patients over 50 years of age and systemically healthy patients who needed an anterior mandibular tooth /teeth extraction for periodontal reasons. All teeth were healthy with mobility grade 3, without infection
3- Dear reviewer
Sterilizing the crushed dentin particles with an autoclave usually influence the morphology of collagen in the teeth. Ultraviolet disinfection would be better.
The best disinfection method is Ultraviolet disinfection, but in this case, we used the autoclave, and I will use your method in the next paper I hope that you can read my new research. Thanks a lot for your comments they were so so necessary for my paper.
It is also true, as you say, that the properties of collagen, growth factors, and morphogenetic proteins are lost, which shows us that the tooth per se is a great independent bone filler material if it is immaturely extracted or sterilized.
4- The standard used for the number of dental particles implanted during surgery is missing.
Biomaterials
One of the biomaterials used was Bioner Bone (Bioner Sistemas Implantológicos, Barcelona, Spain), a synthetic material with a high resorption capacity that allows nice and stable bone regeneration. This material comprises an entirely synthetic bone graft material composed of 100% beta-tricalcium phosphate and features 80% interconnected microporosity and macroporosity. This biomaterial has a granule size ranging from 0.5 to 1 mm particle thickness. (Fig. 1).
Figure 1. Bioner boneÒ (Btcp biomaterial)
Crushed teeth
The study protocol was approved by the Catholic University of Murcia Ethics Committee related to the use of extraction teeth for in-vitro studies (UCAM; registration number 6781; 21-07-2017). Mandibular anterior human teeth were extracted from 10 patients between 50- and 60 years mean, and S.D. was (55.34 ± 0.16) a total including seven men and three women in the study. All the patients signed informed consent forms to donate their teeth for the study and did not receive financial compensation. The patient's teeth were diagnosed with periodontal disease in the lower six anterior teeth (Numbers 31 to 41). The inclusion criteria were patients over 50 years of age and systemically healthy patients who needed an anterior mandibular tooth /teeth extraction for periodontal reasons. All teeth were healthy with mobility grade 3, without infection. Exclusion criteria include patients who do not need an anterior mandibular tooth extraction, systemically complicated patients (ASA III, IV, V), pregnancy or lactation, and patients who unwillingness to follow-up calls. The inclusion and exclusion criteria were described in a previous article by Calvo Guirado et al. [14]. After humans extracted teeth were cut, the crown used only the root. The roots were cleaned with a carbide-tungsten bur then a dental chair syringe dried the roots and ground them using a designed 'Smart Dentin Grinder Genesis' device. The 300-1200 um dentin particulate was sieved through a particular sorting system. (Fig. 2). The standard used for the number of dental particles implanted during surgery was 0.5-1 mm particle thickness
Is the implantation volume comparable to BTCP?
Yes it was the same
Fig. 1-3 should be deleted. Orther figures should be integrated into one figure.
Dera reviewer
Figures 1-3 there are so important for the explanation of using crushed teeth and the type of particles that we can obtain after grinding.
Dear Reviewer
We integrated the surgical procedure into one figure many thanks.

Reviewer 3 Report
Abstract
please change empty and unfilled to non-grafted through out the text
52-54 please rewrite for clarity
110 please delete this statement
185 please provide a citation for this statement
197 please rewrite the run-on sentence
422 please provide a citation for this statement: osteo-inductivity of dentin is doubtful
Please rewrite this important manuscript with a person fluent in English.
This manuscript needs to be rewritten for clarity and ease of reading.
Author Response
please change empty and unfilled to non-grafted throughout the text
52-54 please rewrite for clarity
Dear Reviewer
We corrected all that suggested in the paper for clarity
The use of collagen membranes is intended to seal the soft tissue and allow hard tissue to grow under it without connective tissue. [2]. Many times part of this regeneration is not achieved because the defect is critical, is greater than 6 mm and bone formation is not obtained correctly, increasing bone resorption, added to this, the lengthening of the postoperative time to be able to place the implants. [3].
110 please delete this statement
Dear Reviewer
We deleted your statement that suggested
185 please provide a citation for this statement
Dear Reviewer
DONE
Figure 4. Scanning electron microscopy (S.E.M.) of human crushed sterilized tooth particles.
197 Please rewrite the run-on sentence
The development of this study of critical effects allows the rabbit to be used as an experimental model to regenerate bone quickly and effectively. The critical effects are difficult to achieve, but in these experimental animals it allows us to regenerate the bone more quickly and we can evaluate in short periods of time what happens in the human being.
422 please provide a citation for this statement: osteoinductivity of dentin is doubtful
In addition, bovine dentin has osteoconductive and non-osteoinductive potential, like human dentin, which has been used to repair and promote regeneration in bone tissue [25].
Please rewrite this important manuscript with a person fluent in English.
Dear reviewer the revision was completely done
Many thanks for your comments
